# Biomarker Changes in Response to a 12-Week Supplementation of an Oral Nutritional Supplement Enriched with Protein, Vitamin D and HMB in Malnourished Community Dwelling Older Adults with Sarcopenia

**DOI:** 10.3390/nu14061196

**Published:** 2022-03-11

**Authors:** Suzette L. Pereira, Marni E. Shoemaker, Susan Gawel, Gerard J. Davis, Menghua Luo, Vikkie A. Mustad, Joel T. Cramer

**Affiliations:** 1Abbott Nutrition, 3300 Stelzer Road, Columbus, OH 43219, USA; menghua.luo@abbott.com; 2College of Health Sciences, The University of Texas at El Paso, El Paso, TX 79968, USA; meshoemaker@miners.utep.edu (M.E.S.); jtcramer@utep.edu (J.T.C.); 3Abbott Diagnostics, 100 Abbott Park Road, Abbott Park, Chicago, IL 60064, USA; susan.gawel@abbott.com (S.G.); gerard.davis@abbott.com (G.J.D.); 4Nutrition Science Consulting, LLC, Galena, OH 43021, USA; vmustad51@gmail.com

**Keywords:** sarcopenia, malnutrition, oral nutrition supplementation, biomarkers, HMB

## Abstract

Malnutrition and sarcopenia commonly overlap and contribute to adverse health outcomes. Previously, chronic supplementation with two oral nutritional supplements (ONS), control (C_ONS_) and experimental ONS enriched with protein, vitamin D and β-hydroxy β-methylbutyrate (HMB) (E_ONS_), improved muscle strength and quality in malnourished sarcopenic older adults, with E_ONS_ demonstrating early strength benefits at 12 weeks. To understand the underlying biological mechanisms contributing to the observed early strength benefits of E_ONS_, we examined serum biomarker changes in response to 12-week supplementation. Serum samples (E_ONS_ (*n* = 90) and C_ONS_ (*n* = 103)) collected at baseline and 12 weeks were analyzed. Biomarkers (*n* = 243) were measured using multiplexed immunoassay, commercial immunoassays and ELISAs. Sixty markers were excluded with levels below assay detection limits. Sixteen biomarkers significantly changed in response to both interventions including nutritional and metabolic markers. Thirteen biomarkers significantly changed in response to E_ONS_ but not C_ONS_. Increases in immunoglobulins, myoglobin, total protein, vitamin E and magnesium were observed with E_ONS_. Inflammation-related ferritin and osteopontin decreased, while soluble receptors for cytokines increased, suggesting decreased inflammation. Sex hormone-binding globulin associated with sarcopenia also decreased with E_ONS._ Biomarkers reflective of multiple biological systems were impacted by nutritional intervention in sarcopenic older adults. Incremental biomarker changes were observed in response to E_ONS_ containing HMB that possibly link to improvements in skeletal muscle health.

## 1. Introduction

Malnutrition and sarcopenia are two conditions that can occur simultaneously and are particularly prevalent in older adults. Malnutrition plays a major role in the development of sarcopenia [1], with muscle loss now considered an important characteristic of malnutrition [2] and has been included in the recent consensus definition of malnutrition [3]. Malnutrition and sarcopenia independently, and in conjunction, contribute to an increased risk of adverse health events such as reduced quality of life, mobility disability, hospitalization and mortality [4,5,6,7]. Thus, there is a need for interventions that can both restore nutrition status, as well as address loss of muscle strength and function towards improving quality of life, physical functionality, and long-term well-being.

Recently, chronic nutritional intervention studies using specific nutrients that target muscle health have shown positive benefits on muscle strength, quality and function in sarcopenic populations [8,9,10]. Protein has well-known anabolic effects on muscle and low protein intake has been associated with many negative health outcomes [11,12]. Thus, older adults are recommended to consume at least 1.0–1.2 g protein/kg body weight/day, with even higher intake levels recommended for malnourished patients [13,14].

Vitamin D_3_ supplementation has been demonstrated to improve bone and muscle health, and prevent falls and fractures leading to mobility disability [15,16]. Supplementation is especially relevant in malnourished older adults who have reduced sunlight exposure, and a decrease in synthesis capacity of skin leading to vitamin D deficiency [17,18].

β-hydroxy β-methylbutyrate (HMB), a metabolite of leucine with anabolic properties, has been evaluated in numerous studies as an intervention for improving muscle health outcomes especially in populations at risk of muscle loss [19]. In addition, in healthy older adults, HMB has been shown to preserve muscle during extended bed rest [13] and improve muscle mass and strength in the absence of exercise training [20,21]. Mechanistically, HMB can impact multiple pathways associated with muscle metabolism. It has been shown to stimulate muscle protein synthesis via the activation of the mechanistic target of the rapamycin (mTOR) system, the classical regulator of muscle anabolism, and also stimulate the growth hormone/insulin-like growth hormone factor (IGF-1) axis [22,23,24]. It has also been shown to downregulate muscle protein breakdown via the ubiquitin proteasome pathway and the lysosomal autophagy pathway [25,26]. Animal studies on muscle wasting demonstrated that HMB downregulates expression of Foxhead Box O3 (FOXO3) and Nuclear Factor-kappa B (NF-κB), classical mediators of inflammation involved in muscle wasting [27,28].

Older adults with malnutrition and sarcopenia may not consume sufficient amounts of high-quality protein, vitamin D and/or other nutrients targeting muscle health through meals alone. Oral nutritional supplements (ONS) that provide high-quality protein, vitamins, minerals and calories are recommended to meet the basic nutritional needs of such individuals when diet alone is not enough [29]. Furthermore, specialized ONS containing HMB, designed to address muscle loss, have been explored in several studies involving malnourished individuals and have shown beneficial effects on nutritional status, muscle strength, quality of life, and activities of daily living [30,31,32]. However, there is a gap in understanding the biological systems that are impacted by specialized ONS containing HMB, to help in understanding the mechanisms behind its clinical benefits in malnourished adults. Blood biomarker analysis provides an opportunity to gain such insights into the mechanistic pathways modulated by an intervention, which could possibly lead to expanding the application of the intervention to new populations. Towards this end, we selected one of our previous ONS intervention studies in malnourished sarcopenic older adults, from which blood samples were available for biomarker analysis [8]. In this previous study, compared to a control ONS (C_ONS_) that provided protein and vitamin D, supplementation with a specialized ONS enriched with higher protein, higher vitamin D and HMB (E_ONS_) resulted in early improvements (within 12 weeks) in strength and muscle quality in sarcopenic subpopulations [8]. We used a non-targeted biomarker approach to comprehensively explore physiological pathways that are responsive to nutritional intervention that may have contributed to these muscle health and other systemic benefits. The purpose of this current study was to determine if the E_ONS_ would modulate additional biomarkers beyond those modulated by C_ONS_ in malnourished sarcopenic older adults at the end of a 12-week intervention period.

## 2. Materials and Methods

### 2.1. Participants

In our previous prospective, randomized, double-blinded, controlled study [8], 330 older adults (125 males/205 females) (median age of 77 years) were included from eight countries across Europe and North America. Participants were included if they were (1) at risk or had malnutrition determined by the Subjective Global Assessment (SGA) [33], and (2) had sarcopenia according to the European Working Group on Sarcopenia in Older People (EWGSOP 1) definition [34]. Sarcopenia was defined as low grip strength (<20 kg in females; <30 kg in males) and/or low gait speed (<0.8 m·s^−1^) combined with low skeletal muscle index [34]. Participants were randomized into one of two groups: control ONS (C_ONS_) or experimental ONS (E_ONS_). Detailed information on this study protocol and inclusion/exclusion criteria were reported previously [8]. The original study protocol (previously published; Cramer et al., 2016) [8] was reviewed by local ethics committees or institutional review boards of all European locations and by Quorum IRB (Approval #25390) for the U.S. sites and was conducted according to the guidelines of the Declaration of Helsinki. All participants signed a written informed consent. This study was registered on ClinicalTrials.gov, NCT01191125. All IRB protocols at their respective institutions were approved between 2010 and 2012, with the last amendment to the protocols made in 2012.

### 2.2. Study Design

Over the duration of this study, participants were instructed to drink two servings of C_ONS_ or E_ONS_ products daily between regular meals for 24 weeks as described in Cramer et al. [8]. The blinded products were isocaloric and provided 330 kcals per serving. Each serving of the C_ONS_ (Ensure Plus; Abbott, Zwolle, The Netherlands) contained 14 g protein, 11 g fat, 44 g carbohydrate, 147 IU vitamin D_3_, and additional vitamins and minerals. Each serving of the E_ONS_ (Ensure Plus Advance, Abbott, Zwolle, Netherlands) contained 20 g protein, 11 g fat, 36 g carbohydrate, 1.5 g Ca-HMB, 499 IU vitamin D_3_, and additional vitamins and minerals. Participants were instructed to continue their usual diet, physical activity, and lifestyle habits, along with additional instructions to consume the study product daily and to adhere to a recommended meal plan containing 0.8 g·kg^−1^·d^−1^ of protein at a minimum.

For the analysis of blood biomarkers, fasted serum samples that were collected at baseline and at 12 weeks post-intervention were used. A total of 193 subjects (E_ONS_ (*n* = 90) and C_ONS_ (*n* = 103)) met the requirement for having fasted blood samples collected at both timepoints (baseline and 12 weeks) and were included in the analysis. Biomarkers that were analyzed included an extensive range of biomarker categories, such as inflammatory markers, immune markers, metabolic markers, hormones, nutritional markers, cytokines and growth factors; many of these are linked to muscle/metabolism based on existing literature. A total of 190 biomarkers were measured using the multiplexed immunoassay array Human DiscoveryMap ^®^ 175+ v1.0 (Myriad-RBM, Austin, TX, USA) (Appendix A), of which 60 biomarkers were excluded from evaluation due to results being below assay detection levels in ≥30% of subjects (Appendix A). In addition, 47 clinically approved Invitro Diagnostic Tests markers (ICON, Framingdale, NY, USA) were measured. Other markers included were: (1) estradiol, sex hormone-binding globulin (SHBG), and dehydroepiandrosterone (DHEA) measured on ARCHITECT (Abbott, Chicago, IL, USA), and (2) commercially available enzyme-linked immunoassay (ELISA) kits for plasma total C-terminal agrin fragment (tCAF) (Neurotune AG, Schlieren, Switzerland), insulin-like growth factor-1 (IGF-1) (R&D systems, Minneapolis, MN, USA), and vitamin E (MyBiosource Inc., San Diego, CA, USA).

### 2.3. Statistical Analyses

Baseline values were expressed as the median and interquartile ranges (IQR) for continuous variables and as percentages for categorical variables. Biomarker means ± SD and mean percent change ± SD were calculated and separate paired-sample *t*-tests were performed to compare change across time (baseline to 12 weeks) for each group separately. Sidak-adjusted *p*-values were calculated to account for the number of simultaneous tests. A Pearson product-moment correlation analysis was performed to examine associations between selected metabolic and muscle-related baseline markers and measurements of skeletal muscle mass and strength. All statistical analyses were performed with Microsoft Excel, version 16.10 or SAS 9.4 (SAS Institute, Cary, NC, USA).

## 3. Results

From the original 330 participants, a total of 193 participants (75 males, 118 females) were included in this analysis, based on having fasted blood samples collected at baseline and 12 weeks post-intervention. Baseline characteristics of the included participants are shown in Table 1. There were no significant differences between groups at baseline. The median age was 77 years, and 61.1% were females. A majority of participants were non-obese (79.3%) and were considered normal weight or overweight when categorized by body mass index (BMI) (32.1% and 43.5%, respectively) (Table 1). As previously described, all included participants were categorized with a Subjective Global Assessment (SGA) rating of “B” (mild to moderate malnourishment) and had low skeletal muscle mass [8]. Overall, 73.1% had low handgrip strength, with 60.0% of males and 81.4% of females classified as having low grip strength as defined by the EWGSOP 1 criteria (<30 kg in males, <20 kg in females). Low gait speed was indicated in 40.4% of the participants.

Sixteen serum biomarkers were found to change significantly from baseline in both intervention groups at the end of the 12-week period (Table 2). Six of these were nutritional biomarkers including pre-albumin, transferrin, vitamin B12, blood urea nitrogen (BUN), apolipoprotein C III (Apo C-III) and apolipoprotein (a) (LP(a)), all of which showed a significant increase from their baseline values. Metabolic markers including insulin-like growth factor-1 (IGF-1) and leptin increased in response to the nutritional interventions and levels of IGF binding protein-2 (IGFBP-2) decreased (Table 2). 

The E_ONS_ group but not the C_ONS_ group displayed significant changes in 12 additional biomarkers (Figure 1). Upon 12 weeks of supplementation with E_ONS_, ferritin and osteopontin decreased from baseline by 17.73% and 18.24%, respectively, and IL-6 receptor (IL-6r), TNF-α receptor 1 and 2 (TNFR1 and TNFR2) increased by 4.29%, 5.04%, and 6.48%, respectively. Immunity-related markers such as immunoglobulin A (IgA) and immunoglobulin M (IgM) increased by 6.46% and 11.40% from baseline, respectively. Magnesium, total protein, and vitamin E showed significant increases from baseline of 3.35%, 1.86%, and 15.15%, respectively. Sex hormone-binding globulin (SHBG) decreased by 11.59% and myoglobin increased by 13.02%. Means ± standard deviation (SD) and mean percent change for both E_ONS_ and C_ONS_ of these reported biomarkers can be found in Appendix A.

## 4. Discussion

Our previous work has shown the benefits of a 24-week intervention with two high-quality nutritional supplements on improving muscle strength and functional outcomes in malnourished older adults with sarcopenia [8]. Both nutritional supplements (C_ONS_ and E_ONS_) provided macronutrients including protein, calories, and key micronutrients to address malnutrition. Additionally, E_ONS_ contained HMB, a leucine metabolite with known muscle health benefits, [8,19] in addition to higher protein and higher vitamin D than C_ONS_. In sarcopenic subpopulations, 12 weeks of intervention with E_ONS_ resulted in significant improvements in leg strength and muscle quality, compared to C_ONS_, indicating additional benefits of E_ONS_ [8]. In this study, we wanted to understand the underlying biochemical and physiological changes that occur in response to nutritional supplementation in malnourished sarcopenic older adults, and also to determine if there were differences in biomarkers in response to the two different supplements. A comprehensive biomarker approach was utilized to look at changes in a vast array of serum biomarkers, many of which have been previously associated with muscle mass, strength, and metabolism. We intentionally used a broad screening approach since limited data exist on biomarker changes in response to nutrition in sarcopenic malnourished older adults.

There were several shared biomarkers that changed in response to the two ONS interventions (Table 2), indicating that these biomarkers were responsive to nutrition in general. Lower levels of prealbumin, transferrin, IGF-1 and leptin are known to be associated with malnutrition [35,36,37] and as expected, levels of these markers increased following nutritional intervention. IGF-1 is also a well-known anabolic factor associated with muscle strength and performance [38,39]. Elevated IGFBP-2 was recently found to be associated with malnutrition [40], and the observed decrease in IGFBP-2 levels following nutritional intervention is potentially reflective of an improved nutrition status. Additionally, observed increases in apolipoprotein levels, BUN, vitamin B12, and vitamin D are be tied to the delivery of dietary protein, lipids and vitamins by the nutritional supplements.

Other markers that increased in response to both ONS treatments have not been previously reported within the context of malnutrition. Clusterin is a protein reported to be involved in cellular lipid metabolism and also known to bind leptin and ghrelin [41], which could explain the increase in clusterin following ONS intervention. Interleukin-2 receptor-α (IL-2ra), the receptor for IL-2 found on T lymphocytes, also significantly increased upon ONS intervention. IL2- receptors are upregulated in response to IL-2 produced predominantly by T lymphocytes [42]. IL-2 has been shown to decrease in malnourished individuals [43], tied to alterations in immune status. Although we could not measure changes in IL-2 levels due to the lower limit of quantitation of the assay, it is possible that an increase in IL-2ra indicates a general improvement in T-cell health parallel to an improvement in nutritional status. Epidermal growth factor receptor (EGFR) is a tyrosine kinase receptor expressed in multiple organs and plays important roles in proliferation, survival, and differentiation [44]. EGFR plays a critical role in gastrointestinal tract homeostasis, and it’s associated signaling pathway has been previously shown to be activated by certain amino acids [45]. In addition, EGFR activation is linked to gut hormone signaling (G protein-coupled peptide YY neuropeptide and GLP-2) [44] that could explain its response to changes in nutrition status. Thrombomodulin acts as an anticoagulant and also in suppressing inflammation [46], indicating that an increase in concentration following nutritional supplementation could signal an improvement in health status. Systemic complement C3 and serum amyloid P-component (SAP) regulate many aspects of innate immunity, and modest increases in their levels possibly reflect improvements in immune health status following nutritional supplementation [47,48]. Complement C3 (C3) is also known to be responsive to dietary modulation which could explain its increase over the nutritional intervention period [49]. The relevance of the endothelial adhesion molecule, e-selectin, is unknown since levels of other endothelial adhesion molecules did not increase.

Thirteen additional biomarkers significantly changed in response to the E_ONS_ intervention but not in the C_ONS_ group (Figure 1). This could be related to the compositional differences between the two supplements, with the E_ONS_ group containing HMB, as well as higher protein and higher vitamin D. Myoglobin was found to increase with E_ONS_ intervention. Myoglobin has a functional role in oxygen delivery to muscle important for muscle adaptations. Weber et al. [50] found myoglobin levels to be 48% lower in patients with cancer cachexia compared to healthy controls. Additionally, myoglobin was positively related to quadriceps cross-sectional area (CSA) in both healthy individuals and patients with cachexia. Our previous work has shown that myoglobin levels increase in response to HMB supplementation in older adults on bed rest [51]. Thus, the observed increase in myoglobin levels could be reflective of positive skeletal muscle adaptations in response to HMB in the E_ONS_.

Older adults are known to exhibit declines in both the innate and adaptive immune systems, resulting in increased susceptibility to- and severity of infections [52,53,54]. Beyond the negative effects of aging, malnutrition is known to negatively impact the hematopoietic and lymphoid organs, compromising immunity [55,56]. A reduction in IgA secretion can lead to increased risk of mucosal infections, especially in malnourished individuals [57]. Chronic intervention with E_ONS_ resulted in significant increases in IgA and IgM levels, indicating improvements in overall immune health status in response to E_ONS_. It is possible that HMB along with protein and vitamin D could have played a role in modulating these immune markers. Vitamin D is well recognized as an immune modulator [58]. A positive benefit of HMB on immunity has been previously demonstrated in animals [59,60,61]. Along these lines, another nutritional intervention study reported an increase in IgM levels in response to ONS containingHMB in hospitalized older patients with comorbidities [62].

Other nutritional markers such as total protein, magnesium and vitamin E levels also significantly increased in the E_ONS_ group only, indicating improved nutritional status, tied to the differences in composition of E_ONS_ and C_ONS_ [8]. Vitamin E has been shown to be associated with skeletal muscle health. Semba et al. [63] reported that higher circulating vitamin E levels were associated with muscle strength measurements including grip strength and leg extension strength in sarcopenic older females [63]. Magnesium levels (within normal range) have been reported to have an association with muscle performance, specifically in older adults [64]. Specifically, Domingueze et al. [64] reported associations between magnesium concentrations and measurements of strength and power in older adults [64]. Serum vitamin D was previously reported to significantly increase from baseline in both groups, with the E_ONS_ group showing a significantly greater increase compared to the C_ONS_ group [8]. Additionally, high sex hormone-binding globulin (SHBG) levels have been associated with sarcopenia in both older men and women [65] and was found to decrease in response to E_ONS_ intervention. This is possibly related to the higher vitamin D doses delivered by E_ONS_ (1000 IU·d^−1^), since there appears to be an inverse relationship between Vitamin D and SHBG tied to muscle health [66].

Sub-acute inflammation manifested by small but significant increases in levels of inflammatory biomarkers (e.g., IL-6 and TNF-α) has been reported with aging and sarcopenia [67,68]. In the present study, baseline levels of IL-6 and TNF-α were below the lowest limit of quantitation of the assays and thus excluded from analysis. It appears that the assays for IL-6 and TNF-α present on the multiplex we employed had lower sensitivity than those previously reported. For example, Visser et al. [67] reported IL-6 levels > 1.8 pg·mL^−1^ to be associated with low muscle mass, and >80% of their population had IL-6 levels ≤ 4 pg·mL^−1^, whereas the lowest detectable level of IL-6 in our assay was 4.2 pg·mL^−1^ [67].

However, there was some evidence for sub-acute inflammation in our study participants based on changes in osteopontin, ferritin, and cytokine receptors. Osteopontin is a protein well associated with inflammation and other disease states [69]. Karadag et al. [70] reported an association between high osteopontin levels and weight loss in patients with cancer [70]. In the present study, E_ONS_ supplementation resulted in an 18% decrease in osteopontin levels, implying a decrease in inflammation. In addition, supplementation with E_ONS_ resulted in an increase in levels of the soluble receptors for IL-6 and TNF-α. These soluble receptors have been reported to have a neutralizing effect on their respective cytokines. The soluble IL-6 receptor has previously been reported to act as a buffer by binding circulating IL-6, thus neutralizing its inflammatory effects [71]. Lustosa et al. [72] reported higher levels of soluble receptor for TNF-α (TNFR1) in non-sarcopenic women compared to sarcopenic women [72]. Therefore, it is possible that higher levels of circulating TNFR1 could bind TNF-α and hamper its related inflammatory cascade in non-sarcopenic individuals, protecting skeletal muscle [72,73]. Ferritin levels also decreased by ~18% in response to E_ONS_. Although ferritin is a marker of iron deficiency, increases in serum ferritin levels have been reported to occur during age-associated inflammation, even in subjects who were not iron deficient [74]. These findings point to an anti-inflammatory benefit of E_ONS_ which could help explain the early improvements in muscle strength observed at the 12 week timepoint. Previous studies have shown that HMB can modulate levels of circulating inflammatory cytokines in populations under physical stress [75,76]. A recent study in older adults with sarcopenia demonstrated that compared to exercise alone, a 12-week intervention using ONS with HMB along with exercise could modify T cell-specific inflammatory gene expression, and was associated with improved lower limb muscle strength performance [77].

A limitation to our study is that a subgroup of the original study participants (193 out of a total of 330 participants) were used for biomarker analysis based on availability of blood samples. However, the subgroups analyzed were still relatively large in size and well matched for their baseline characteristics. Another limitation is that we were not able to measure the extremely low sub-clinical levels of some inflammatory markers. However, we were able to look at alternate inflammation-related markers to draw conclusions. One additional limitation is that because the E_ONS_ supplement contained higher levels of protein and vitamin D in addition HMB compared to the C_ONS_ supplement, it is not possible to isolate which ingredient in E_ONS_ was directly responsible for the observed biomarker changes. It is possible that the effects could be due to a synergy between the various components in E_ONS_.

## 5. Conclusions

Taken together, the findings from this study indicate that in sarcopenic malnourished older adults, multiple biochemical pathways related to nutrition, immunity, inflammation, anabolism and muscle metabolism can be impacted by nutrition. Many of these markers are known to be associated with muscle health outcomes. In our study, correlation analysis (Appendix A) revealed that baseline levels of markers such as IGF-1 and prealbumin positively correlated with muscle strength, whereas markers such as IGFBP-2, leptin, osteopontin, and SHBG negatively correlated with muscle strength. These data also show that it is possible to modulate pathways linked to improving muscle metabolism using a high protein, high vitamin D oral nutrition supplement containing HMB.

It is possible that some of the biomarkers reported herein could be useful surrogates to track/monitor progression of sarcopenia, or could also be early indicators of muscle health status in people at risk of malnutrition and sarcopenia. These exploratory findings will need to be validated in additional intervention studies in sarcopenic malnourished older adults. Additionally, further investigative studies are needed to explore the connection between immunity, inflammation and muscle metabolism. This will aid in developing a more targeted approach to address sarcopenia development and progression.

## Figures and Tables

**Figure 1 nutrients-14-01196-f001:**
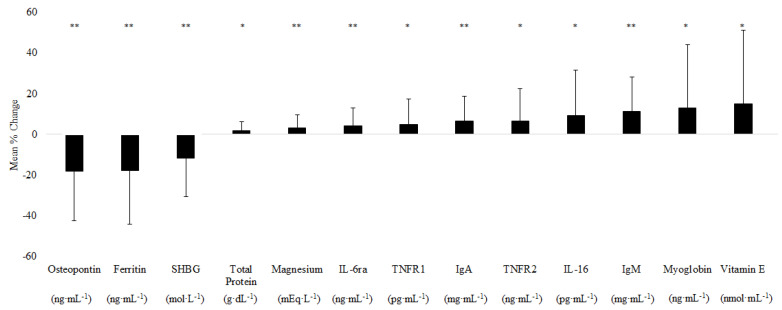
Biomarker changes specific to experimental ONS (E_ONS_) following 12-week supplementation. Values are the means ± SD, compared by univariable dependent *t*-test with Sidak-adjusted *p*-value < 0.05. Interleukin-6 receptor-α (IL-6ra), tumor necrosis factor receptor-1 (TNFR1), tumor necrosis factor receptor-2 (TNFR2), immunoglobulin A (IgA), immunoglobulin M (IgM), and sex hormone-binding globulin (SHBG). * indicates *p* < 0.05; ** indicates *p* < 0.01.

**Table 1 nutrients-14-01196-t001:** Baseline characteristics of study population used for biomarker analysis.

		Overall (*n* = 193)	E_ONS_ (*n* = 90)	C_ONS_ (*n* = 103)	*p*-Value ^c^
Age (years) ^a^		77 (71, 81)	77.5 (71, 82)	76 (71, 81)	0.308
Sex ^b^	Males	75 (38.9%)	36 (40.0%)	39 (37.9%)	0.761
	Females	118 (61.1%)	54 (60.0%)	64 (62.1%)	
Obese (kg·m^−2^) ^b^	Obese ≥ 30	40 (20.7%)	18 (20.0%)	22 (21.4%)	0.816
	Non-Obese < 30	153 (79.3%)	72 (80.0%)	81 (78.6%)	
Body Mass Index ^a^		26.7 (23.5, 29.1)	26.9 (23.1, 29.0)	26.3 (23.9, 29.2)	0.882
<18.5	Low ^b^	7 (3.6%)	3 (3.3%)	4 (3.9%)	0.995
18.5–24.9	Normal ^b^	62 (32.1%)	29 (32.2%)	33 (32.0%)	
25.0–29.9	Overweight ^b^	84 (43.5%)	40 (44.4%)	44 (42.7%)	
≥30.0	Obese ^b^	40 (20.7%)	18 (20.0%)	22 (21.4%)	
Percent Total Lean Mass (%) ^a^		58.7 (54.3, 66.7)	59.2 (55.3, 66.3)	58.3 (53.5, 67.4)	0.165
Percent Leg Lean Mass (%) ^a^		30.4 (28.7, 32.0)	30.4 (29.17, 32.0)	30.4 (28.5, 32.0)	0.846
Handgrip Strength (kg) ^a^		18.8 (15.0, 27.3)	19.2 (15.3, 29.3)	18.7 (14.0, 26.0)	0.285
	Low ^b^	141 (73.1%)	60 (66.7%)	81 (78.6%)	0.061
	Normal ^b^	52 (26.9%)	30 (33.3%)	22 (21.4%)	
Male Handgrip Strength ^a^		27.8 (22.7, 34.7)	29.6 (22.5, 34.7)	26 (22.7, 33.0)	0.411
	Low ^b^	45 (60.0%)	19 (52.8%)	26 (66.7%)	0.220
	Normal ^b^	30 (40.0%)	17 (47.2%)	13 (33.3%)	
Female HandgripStrength ^a^		16.6 (13.2, 19.0)	17 (14.0, 19.5)	16.17 (12.7, 18.7)	0.384
	Low ^b^	96 (81.4%)	41 (75.9%)	55 (85.9%)	0.164
	Normal ^b^	22 (18.6%)	13 (24.1%)	9 (14.1%)	
Average Extensor Peak Torque (Nm) ^a^		58.7 (37.1, 78.3)	58.8 (36.3, 79.2)	58.7 (37.1, 78.3)	0.753
Peak Extensor Peak Torque (Nm) ^a^		62.2 (42.0, 82.0)	62.2 (39.4, 81.1)	62.9 (45.0, 82.7)	0.841
Gait Speed (m·s^−1^) ^b^	Low	115 (59.6%)	57 (63.3%)	58 (56.3%)	0.321
	Normal	78 (40.4%)	33 (36.7%)	45 (43.7%)	

^a^ Values are the median (25th, 75th interquartile range). ^b^ Values are the number of participants (percentages (%)). ^c^ Differences between treatment groups using independent-samples *t*-tests, *p* < 0.05.

**Table 2 nutrients-14-01196-t002:** Biomarkers common to both treatment groups (E_ONS_ and C_ONS_) that significantly changed over 12 weeks of nutritional supplementation.

	E_ONS_ (*n* = 90) ^a^			C_ONS_ (*n* = 103)		
Biomarker	Baseline	12 Weeks	Percent Change	Baseline	12 Weeks	Percent Change
Apolipoprotein(a) (Lp(a) (μg·mL^−1^)	309.66 ± 305.59	388.60 ± 406.06	22.11 ± 31.72 ^c^	314.57 ± 356.34	339.66 ± 378.41	14.16 ± 36.51 ^b^
ApolipoproteinC III (Apo C-III) (μg·mL^−1^)	229.32 ± 83.82	243.11 ± 80.41	8.60 ± 18.98 ^b^	229.01 ± 75.94	249.51 ± 82.57	10.49 ± 21.11 ^c^
Blood Urea Nitrogen (BUN) (mg·dL^−1^)	18.71 ± 6.25	24.52 ± 9.31	33.86 ± 32.06 ^c^	19.37 ± 7.50	21.18 ± 7.73	13.29 ± 24.49 ^c^
Pre-Albumin (mg·dL^−1^)	24.32 ± 5.27	25.60 ± 5.43	6.96 ± 16.15 ^b^	24.31 ± 5.20	25.69 ± 4.91	7.18 ± 15.17 ^c^
Transferrin (mg·dL^−1^)	219.69 ± 42.68	239.10 ± 45.38	9.74 ± 13.47 ^c^	227.37 ± 48.57	241.74 ± 52.05	6.85 ± 11.51 ^c^
Vitamin B12 (mg·dL^−1^)	593.76 ± 267.79	647.03 ± 273.07	14.80 ± 36.08 ^b^	529.26 ± 287.80	600.35 ± 293.74	21.27 ± 37.06 ^c^
Clusterin (μg·mL^−1^)	215.67 ± 29.84	225.02 ± 31.48	4.81 ± 10.80 ^b^	209.23 ± 27.35	219.58 ± 29.63	5.47 ± 10.80 ^c^
Complement C3 (mg·mL^−1^)	1.09 ± 0.21	1.15 ± 0.22	6.41 ± 15.50 ^b^	1.13 ± 0.22	1.20 ± 0.23	6.43 ± 11.85 ^c^
Epidermal Growth Factor Receptor (EGFR) (ng·mL^−1^)	3.51 ± 0.55	3.83 ± 0.67	9.50 ± 11.80 ^c^	3.50 ± 0.53	3.68 ± 0.57	5.50 ± 10.42 ^c^
E-Selectin (ng·mL^−1^)	8.65 ± 3.71	9.35 ± 3.70	9.89 ± 14.31 ^c^	9.67 ± 4.94	10.55 ± 4.83	10.89 ± 16.36 ^c^
Interluekin-2 Receptor Alpha (IL-2ra) (pg·mL^−1^)	2582.33 ± 1590.01	2711.889 ± 1617.25	6.23 ± 14.94 ^b^	2554.37 ± 1094.80	2683.98 ± 1131.11	5.88 ± 14.65 ^b^
Serum Amyloid P-Component (SAP) (μg·mL^−1^)	13.90 ± 4.22	15.07 ± 4.30	10.54 ± 18.43 ^c^	13.76 ± 3.65	15.20 ± 3.68	21.48 ± 17.56 ^c^
Thrombomodulin (ng·mL^−1^)	5.51 ± 1.59	5.72 ± 1.49	4.91 ± 11.34 ^b^	5.70 ± 1.82	5.93 ± 1.83	4.72 ± 11.56 ^b^
Insulin-Like Growth Factor-1 (IGF-1) (ng·mL^−1^)	1.76 ± 0.71	2.38 ± 1.01	46.96 ± 77.08 ^c^	1.73 ± 0.73	2.21 ± 0.96	38.39 ± 66.14 ^c^
Insulin-Like Growth Factor Binding Protein-2 (IGFBP-2) (ng·mL^−1^)	170.47 ± 82.00	141.56 ± 74.81	−15.44 ± 15.96 ^c^	151.26 ± 72.34	132.96 ± 71.96	−11.13 ± 22.15 ^c^
Leptin (ng·mL^−1^)	13.78 ± 10.71	18.76 ± 15.38	46.72 ± 55.73 ^c^	18.48 ± 20.44	24.20 ± 22.03	46.49 ± 53.02 ^c^

Values are represented as the means ± standard deviations (SD). ^a^ E_ONS_
*n* = 89 for vitamin B12; *n* = 88 for prealbumin. ^b^ Change from baseline using univariable dependent *t*-test with Sidak-adjusted *p*-value ≤ 0.05. ^c^ Change from baseline using univariable dependent *t*-test with Sidak-adjusted *p*-value ≤ 0.001.

## Data Availability

The data presented in this study are available on request from the corresponding author.

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
