# Peer review of "Biomarker Changes in Response to a 12-Week Supplementation of an Oral Nutritional Supplement Enriched with Protein, Vitamin D and HMB in Malnourished Community Dwelling Older Adults with Sarcopenia"

_nutrients, 2022, doi:10.3390/nu14061196_

Round 1

Reviewer 1 Report

 Pereira and others submitted a manuscript entitled, "Biomarker changes in response to a 12-week supplementation of an oral nutritional supplement with HMB in malnourished community dwelling older adults with sarcopenia."  They examined the effects  two oral nutritional supplements (with and without β-hydroxy β-methylbutyrate, HMB) on serum biomarkers 12-week changes. The authors concluded biomarkers reflective of multiple biological systems were impacted by nutritional intervention in sarcopenic older adults.  This is a well written manuscript and contained information valuable to researchers in the malnutrition and sarcopenia area of study. 

I one major crititism of the experimental Design.  The two treatments not only different in HMB but also protein, CHO, and Vitamin D3 content.  Even though the authors did not statistically compare the two treatments, it is implied that the HMB containing supplement was more efficacious. Could the difference in biomarkers be because of additional protein or Vitamin D3? The authors did dicuss difference in composition in the discussion but it also should be mention as a limitation.

Minor

Why is this biomarker analysis important?  It not clear in the abstract and could be expanded upon in the introduction.

It is mentioned that samples came from a 24-week study.  Why were only 12 week samples used?  Were 24-week samples available? 

What are the implication or usefulness of these biomarkers in malnutrition and sarcopenia research?

Author Response

REVIEWER 1 COMMENTS:

‘Pereira and others submitted a manuscript entitled, "Biomarker changes in response to a 12-week supplementation of an oral nutritional supplement with HMB in malnourished community dwelling older adults with sarcopenia."  They examined the effects two oral nutritional supplements (with and without β-hydroxy β-methylbutyrate, HMB) on serum biomarkers 12-week changes. The authors concluded biomarkers reflective of multiple biological systems were impacted by nutritional intervention in sarcopenic older adults.  This is a well written manuscript and contained information valuable to researchers in the malnutrition and sarcopenia area of study. 

I one major crititism of the experimental Design.  The two treatments not only different in HMB but also protein, CHO, and Vitamin D3 content.  Even though the authors did not statistically compare the two treatments, it is implied that the HMB containing supplement was more efficacious. Could the difference in biomarkers be because of additional protein or Vitamin D3? The authors did discuss difference in composition in the discussion but it also should be mention as a limitation.

Minor

Why is this biomarker analysis important?  It not clear in the abstract and could be expanded upon in the introduction.

It is mentioned that samples came from a 24-week study.  Why were only 12 week samples used?  Were 24-week samples available? 

What are the implication or usefulness of these biomarkers in malnutrition and sarcopenia research?’

RESPONSE TO REVIEWER 1:

We would like to thank the reviewer for the insightful comments and feedback provided to improve the quality of our manuscript and make it relevant to the readers.

  1. Comment: ‘I have one major criticism of the experimental Design.  The two treatments not only different in HMB but also protein, CHO, and Vitamin D3 content.  Even though the authors did not statistically compare the two treatments, it is implied that the HMB containing supplement was more efficacious. Could the difference in biomarkers be because of additional protein or Vitamin D3? The authors did discuss difference in composition in the discussion but it also should be mention as a limitation’.

RESPONSE: We agree with the reviewer that the two treatments differed in levels of protein and Vit D in addition to the presence of HMB.  In our previous publication, we reported early incremental benefits of the high protein-high Vit D-HMB containing supplement (EONS) on muscle strength when compared to the Control supplement. We agree that the clinical benefits as well as the difference is some of the biomarkers identified could well be attributed to the overall composition of the product and not just due to HMB.  

As recommended, we have included this as a limitation in the manuscript:

- Discussion (Line 321-325 added)

One additional limitation is that because the EONS supplement contained higher levels of protein and Vitamin D in addition HMB compared to the CONS, it is not possible to isolate which ingredient in EONS was directly responsible for the observed biomarker changes. It is possible that the effects could be due to a synergy between the various components in EONS.

In addition, to highlight the compositional difference of the EONS, we have changed the description in the title, abstract and throughout the document to highlight that the EONS was enriched in protein and Vitamin D in addition to HMB.

New Title:

Biomarker changes in response to a 12-week supplementation of an oral nutritional supplement enriched in protein, vitamin D and HMB in malnourished community dwelling older adults with sarcopenia.

Abstract (Line 15-18 modified):

Previously, chronic supplementation with two oral nutritional supplements (ONS), control (CONS) and experimental ONS enriched with protein, vitamin D and b-hydroxy b-methylbutyrate (HMB) (EONS), improved muscle strength and quality in malnourished sarcopenic older adults, with EONS demonstrating early strength benefits at 12-weeks.

Introduction (line 84-93 modified):

In this previous study, compared to a control ONS (CONS) that provided protein and vitamin D, supplementation with a specialized ONS enriched with higher protein, higher vitamin D and HMB (EONS) resulted in early improvements (within 12 weeks) in strength and muscle quality in sarcopenic subpopulations [8]. We used a non-targeted biomarker approach to comprehensively explore physiological pathways that are responsive to nutritional intervention that may have contributed to these muscle health and other systemic benefits. The purpose of this current study was to determine if the EONS would modulate additional biomarkers beyond those modulated by CONS in malnourished sarcopenic older adults at the end of a 12-week intervention period.

  1. Comment: ‘Why is this biomarker analysis important?  It not clear in the abstract and could be expanded upon in the introduction.’

RESPONSE: The following sentences have been added to clarify and expand upon the importance of the biomarker analysis.

Abstract (Line 18-20 added):

To understand the underlying biological mechanisms contributing to the observed early strength benefits of EONS, we examined serum biomarkers changes in response to 12-week supplementation.

Introduction (Lines 77-93) has been changed as follows:

However, there is a gap in understanding the biological systems that are impacted by specialized ONS containing HMB, to help in understanding the mechanisms behind its clinical benefits in malnourished adults. Blood biomarkers analysis provides an opportunity to gain such insights into the mechanistic pathways modulated by an intervention, which could possibly lead to expanding the application of the intervention to new populations. Towards this end we selected one of our previous ONS intervention studies in malnourished sarcopenic older adults, from which blood samples were available for biomarker analysis [8]. In this previous study, compared to a control ONS (CONS) that provided protein and vitamin D, supplementation with a specialized ONS enriched with higher protein, higher vitamin D and HMB (EONS) resulted in early improvements (within 12 weeks) in strength and muscle quality in sarcopenic subpopulations [8]. We used a non-targeted biomarker approach to comprehensively explore physiological pathways that are responsive to nutritional intervention that may have contributed to these muscle health and other systemic benefits. The purpose of this current study was to determine if the EONS would modulate additional biomarkers beyond those modulated by CONS in malnourished sarcopenic older adults at the end of a 12-week intervention period.

  1. Comment: ‘It is mentioned that samples came from a 24-week study.  Why were only 12 week samples used?  Were 24-week samples available?’ 

RESPONSE: The 12-week samples were chosen for analysis because at that timepoint there was a significant early improvement in muscle strength and quality in the EONS group compared to the CONS group. It offered an opportunity to understand if there were certain biological pathways that contributed to these early improvements in muscle outcomes, and explore differences between the two interventions. In addition, due to some technical errors during sampling at 24 weeks, there was not enough serum sample available from a large number of subjects to measure all the markers that we set out to evaluate.

  1. Comment: ‘What are the implication or usefulness of these biomarkers in malnutrition and sarcopenia research?’

RESPONSE: Our study has identified that in people with malnutrition and sarcopenia, multiple biomarkers tied to a variety of biochemical pathways change during the course of nutritional intervention.  At baseline, many of these biomarkers also linked with muscle strength. The connection between immunity, inflammation and muscle metabolism warrants further investigation since markers related to these pathways were identified. Some of the identified biomarkers could be explored as markers to track/monitor progression of sarcopenia or could be used as early indicators of muscle health status.  These findings will need further validation in additional studies.

These statements were previously included in the Conclusion section of the manuscript. We have moved the paragraph related to study limitations previously included in the conclusions into the Discussion section. We hope the Conclusion section is more clear and speaks to the implication of the biomarker findings.

Modified Conclusions (Line 328-353):

Taken together, the findings from this study indicate that in sarcopenic malnourished older adults, multiple biochemical pathways related to nutrition, immunity, inflammation, anabolism and muscle metabolism can be impacted by nutrition. Many of these markers are known to be associated with muscle health outcomes.  In our study, correlation analysis (Table S4) revealed that baseline levels of markers such as IGF-1 and prealbumin positively correlated with muscle strength, whereas markers such IGFBP-2, leptin, osteopontin and SHBG negatively correlated with muscle strength. These data also show that it is possible to modulate pathways linked to improving muscle metabolism using high protein- high vitamin D supplements containing HMB.

It is possible that some of the biomarkers reported herein could be useful surrogates to track/monitor progression of sarcopenia or could also be early indicators of muscle health status in people at risk of malnutrition and sarcopenia. These exploratory findings will need to be validated in additional intervention studies in sarcopenic malnourished older adults. Additionally, further investigative studies are needed to explore the connection between immunity, inflammation and muscle metabolism. This will aid in developing a more targeted approach to address sarcopenia development and progression

We thank you for taking the time to review this manuscript and hope we have provided the necessary answers to your comments.

Reviewer 2 Report

In their work Pereira et al have examined the effects of 12-week ONS supplementation with and without HMB on biomarkers changes from serum of malnourished sarcopenic older adults. A huge number of biomarkers were analyzed (205), of which 59 were excluded as their level was below the evaluation limit.

Although both ONS cause the increase of 16 biomarkers mainly related to nutrition, the authors conclude that the ONS containing HMB (Eons) determines a further increase in the variation of 13 other biomarkers.

The work could be of some interest both to counter the muscle catabolism of sarcopenic patients, and as a screening for the variation of numerous serum markers, however I observe a serious limitation in study design that does not allow to accept the work in this form.

As stated by the authors (also in the title), the focus of the work concerns the effect of HMB on change serum biomarkers. However, two ONS (Cons and Eons) did not differ exclusively for the presence or absence of HMB, but were profoundly different from each other also for protein content and vitamin D3.

In fact, Cons contains 14g of protein and 147 IU of vit. D3, while Eons contains 20g of protein, 499 IU of vit D3 and 1.5g of HMB.

Why was it chosen to compare two ONS so profoundly different (proteins + 43%; vit. D3 + 340%)?

Therefore, comparing two ONS (administered twice a day) it is not possible to state that major variations of the biomarkers observed in Eons are due to HMB alone and are not influenced by the excess of proteins and/or vit. D3.

Author Response

REVIEWER 2 COMMENTS:

In their work Pereira et al have examined the effects of 12-week ONS supplementation with and without HMB on biomarkers changes from serum of malnourished sarcopenic older adults. A huge number of biomarkers were analyzed (205), of which 59 were excluded as their level was below the evaluation limit.

Although both ONS cause the increase of 16 biomarkers mainly related to nutrition, the authors conclude that the ONS containing HMB (Eons) determines a further increase in the variation of 13 other biomarkers.

The work could be of some interest both to counter the muscle catabolism of sarcopenic patients, and as a screening for the variation of numerous serum markers, however I observe a serious limitation in study design that does not allow to accept the work in this form.

As stated by the authors (also in the title), the focus of the work concerns the effect of HMB on change serum biomarkers. However, two ONS (Cons and Eons) did not differ exclusively for the presence or absence of HMB, but were profoundly different from each other also for protein content and vitamin D3.

In fact, Cons contains 14g of protein and 147 IU of vit. D3, while Eons contains 20g of protein, 499 IU of vit D3 and 1.5g of HMB.

Why was it chosen to compare two ONS so profoundly different (proteins + 43%; vit. D3 + 340%)?

Therefore, comparing two ONS (administered twice a day) it is not possible to state that major variations of the biomarkers observed in Eons are due to HMB alone and are not influenced by the excess of proteins and/or vit. D3.

RESPONSE TO REVIEWER 2:

We would like to thank the reviewer for the insightful comments and thoughtful feedback provided to help clarify the findings. 

We agree with the reviewer that the two supplements differed not just in HMB content but also in protein and vitamin D levels. It was not our intention to mislead the reader about the nature of the interventions used, and we apologize if that was not very clear.  We did include details of the supplements in the Methods section as well as in the early part of the Discussion (Line 193-209- modified document), however we will clarify in more detail (see below).  

In our original intervention paper (Cramer, J.T.et al. J Am Med Dir Assoc 2016, 17, 1044–1055, doi:10.1016/j.jamda.2016.08.009), we discussed the rationale for comparing these two supplements compositions. We have included statements in the introduction to further expand on the components of the supplements and rationale behind it.  We agree that the effect of the EONS cannot be attributed to HMB alone but rather to the overall supplement which contained higher protein and higher Vitamin D in addition to HMB, compared to the CONS.  In order to further clarify the composition of the EONS supplement, we have made the following modifications to the manuscript:

New Title:

Biomarker changes in response to a 12-week supplementation of an oral nutritional supplement enriched in protein, vitamin D and HMB in malnourished community dwelling older adults with sarcopenia.

Abstract (line 16 modified):

Previously, chronic supplementation with two oral nutritional supplements (ONS), control (CONS) and experimental ONS enriched with protein, vitamin D and b-hydroxy b-methylbutyrate (HMB) (EONS), improved muscle strength and quality in malnourished sarcopenic older adults, with EONS demonstrating early strength benefits at 12-weeks.

Introduction: Line 46-54 added:

Protein has well known anabolic effects on muscle and low protein intake has been associated with many negative health outcomes (Langsetmo et al., 2020; Coelho-Junio et al., 2018). Thus older adults are recommended to consume at least 1.0–1.2 g protein/kg body weight/day, with even higher intake levels recommended for malnourished patients (Deutz et al., 2014; Bauer et all., 2013).

Vitamin D3 supplementation has been demonstrated to improve bone and muscle health, and prevent falls and fractures leading to mobility-disability (Bischoff-Ferrari et al., 2004; Garcia et al., 2019).  Supplementation is especially relevant in malnourished older adults who have reduced sunlight exposure, and a decrease in synthesis capacity of skin leading to vitamin D deficiency (Bruyère  et al., 2014; Bischoff-Ferrari et al., 2012).

References added:

Langsetmo, L.; Harrison, S.; Jonnalagadda, S.; Pereira, S.L.; Shikany, J.M.; Farsijani, S.; Lane, N.E.; Cauley, J.A.; Stone, K.; Cawthon, P.M. Low Protein Intake Irrespective of Source Is Associated with Higher Mortality Among Older Community-Dwelling Men. J Nutr Health Aging 2020, 24, 900–905, doi:10.1007/s12603-020-1422-4.

Coelho-Júnior, H.J.; Rodrigues, B.; Uchida, M.; Marzetti, E. Low Protein Intake Is Associated with Frailty in Older Adults: A Systematic Review and Meta-Analysis of Observational Studies. Nutrients 2018, 10, E1334, doi:10.3390/nu10091334.

Deutz, N.E.P.; Pereira, S.L.; Hays, N.P.; Oliver, J.S.; Edens, N.K.; Evans, C.M.; Wolfe, R.R. Effect of β-Hydroxy-β-Methylbutyrate (HMB) on Lean Body Mass during 10 Days of Bed Rest in Older Adults. Clin Nutr 2013, 32, 704–712, doi:10.1016/j.clnu.2013.02.011.

Bauer, J.; Biolo, G.; Cederholm, T.; Cesari, M.; Cruz-Jentoft, A.J.; Morley, J.E.; Phillips, S.; Sieber, C.; Stehle, P.; Teta, D.; et al. Evidence-Based Recommendations for Optimal Dietary Protein Intake in Older People: A Position Paper from the PROT-AGE Study Group. J Am Med Dir Assoc 2013, 14, 542–559, doi:10.1016/j.jamda.2013.05.021.

Bischoff-Ferrari, H.A.; Dawson-Hughes, B.; Willett, W.C.; Staehelin, H.B.; Bazemore, M.G.; Zee, R.Y.; Wong, J.B. Effect of Vitamin D on Falls: A Meta-Analysis. JAMA 2004, 291, 1999–2006, doi:10.1001/jama.291.16.1999.

Garcia, M.; Seelaender, M.; Sotiropoulos, A.; Coletti, D.; Lancha, A.H. Vitamin D, Muscle Recovery, Sarcopenia, Cachexia, and Muscle Atrophy. Nutrition 2019, 60, 66–69, doi:10.1016/j.nut.2018.09.031.

Bruyère, O.; Cavalier, E.; Souberbielle, J.-C.; Bischoff-Ferrari, H.A.; Beaudart, C.; Buckinx, F.; Reginster, J.-Y.; Rizzoli, R. Effects of Vitamin D in the Elderly Population: Current Status and Perspectives. Arch Public Health 2014, 72, 32, doi:10.1186/2049-3258-72-32.

Bischoff-Ferrari, H.A. “Vitamin D - Why Does It Matter?” - Defining Vitamin D Deficiency and Its Prevalence. Scand J Clin Lab Invest Suppl 2012, 243, 3–6, doi:10.3109/00365513.2012.681938.

Introduction: Line 70-77 added:

Older adults with malnutrition and sarcopenia may not consume sufficient amounts of high-quality protein, vitamin D and/or other nutrients targeting muscle health through meals alone. Oral nutritional supplements (ONS) that provide high quality protein, vitamins, minerals and calories are recommended to meet the basic nutritional needs of such individuals when diet alone is not enough (WHO, 2019). Furthermore, specialized ONS containing HMB, designed to address muscle loss, has been explored in several studies involving malnourished individuals and have shown beneficial effects on nutritional status, muscle strength, quality of life, and activities of daily living (Wilkinson et al., 2013; Eley et al., 2007; Gerlinger-Romero et al., 2011).

References added:

Integrated Care for Older People: Guidelines on Community-Level Interventions to Manage Declines in Intrinsic Capacity; WHO Guidelines Approved by the Guidelines Review Committee; World Health Organization: Geneva, 2017; ISBN 978-92-4-155010-9.

Wilkinson, D.J.; Hossain, T.; Hill, D.S.; Phillips, B.E.; Crossland, H.; Williams, J.; Loughna, P.; Churchward-Venne, T.A.; Breen, L.; Phillips, S.M.; et al. Effects of Leucine and Its Metabolite β-Hydroxy-β-Methylbutyrate on Human Skeletal Muscle Protein Metabolism. J Physiol 2013, 591, 2911–2923, doi:10.1113/jphysiol.2013.253203.

Eley, H.L.; Russell, S.T.; Baxter, J.H.; Mukerji, P.; Tisdale, M.J. Signaling Pathways Initiated by Beta-Hydroxy-Beta-Methylbutyrate to Attenuate the Depression of Protein Synthesis in Skeletal Muscle in Response to Cachectic Stimuli. Am J Physiol Endocrinol Metab 2007, 293, E923-931, doi:10.1152/ajpendo.00314.2007.

Gerlinger-Romero, F.; Guimarães-Ferreira, L.; Giannocco, G.; Nunes, M.T. Chronic Supplementation of Beta-Hydroxy-Beta Methylbutyrate (HMβ) Increases the Activity of the GH/IGF-I Axis and Induces Hyperinsulinemia in Rats. Growth Horm IGF Res 2011, 21, 57–62, doi:10.1016/j.ghir.2010.12.006

Introduction: Line 84-93 modified:

In this previous study, compared to a control ONS (CONS) that provided protein and vitamin D, supplementation with a specialized ONS enriched with higher protein, higher vitamin D and HMB (EONS) resulted in early improvements (within 12 weeks) in strength and muscle quality in sarcopenic subpopulations [8]. We used a non-targeted biomarker approach to comprehensively explore physiological pathways that are responsive to nutritional intervention that may have contributed to these muscle health and other systemic benefits. The purpose of this current study was to determine if the EONS would modulate additional biomarkers beyond those modulated by CONS in malnourished sarcopenic older adults at the end of a 12-week intervention period.

Discussion: Line 262 added:

Vitamin D is well recognized as an immune modulator (Charoenngam et al., 2020)

Charoenngam, N.; Holick, M.F. Immunologic Effects of Vitamin D on Human Health and Disease. Nutrients 2020, 12, 2097. https://doi.org/10.3390/nu12072097

Discussion (Lines 321 – 324 added as a limitation to the study)-

One additional limitation is that because the EONS supplement contained higher levels of protein and vitamin D in addition HMB compared to the CONS, it is not possible to isolate which ingredient in EONS was directly responsible for the observed biomarker changes.  It is possible that the effects could be due to a synergy between the various components in EONS.

We thank you for taking the time to review this manuscript and hope we have provided the necessary answers to address your comments and concerns.

Round 2

Reviewer 2 Report

I thank the authors for making significant additions to the text. In this form I think the comparison between the two proposed diets is more understandable.